# Mutation is all you need

**Lennart Schneider**                                              LENNART.SCHNEIDER@STAT.UNI-MUENCHEN.DE
**Florian Pfisterer**                                              FLORIAN.PFISTERER@STAT.UNI-MUENCHEN.DE
**Martin Binder**                                                  MARTIN.BINDER@STAT.UNI-MUENCHEN.DE
**Bernd Bischl**                                                   BERND.BISCHL@STAT.UNI-MUENCHEN.DE
*Department of Statistics, LMU Munich, Germany*

## Abstract

Neural architecture search (NAS) promises to make deep learning accessible to non-experts by automating architecture engineering of deep neural networks. BANANAS is one state-of-the-art NAS method that is embedded within the Bayesian optimization framework. Recent experimental findings have demonstrated the strong performance of BANANAS on the NAS-Bench-101 benchmark being determined by its path encoding and not its choice of surrogate model. We present experimental results suggesting that the performance of BANANAS on the NAS-Bench-301 benchmark is determined by its acquisition function optimizer, which minimally mutates the incumbent.

## 1. Introduction

Neural architecture search (NAS) methods can be categorized along three dimensions (Elsken et al., 2019a): search space, search strategy, and performance estimation strategy. Focusing on search strategy, popular methods are given by Bayesian optimization (BO, e.g., Bergstra et al. 2013; Domhan et al. 2015; Mendoza et al. 2016; Kandasamy et al. 2018; White et al. 2019), evolutionary methods (e.g., Miller et al. 1989; Liu et al. 2017; Real et al. 2017, 2019; Elsken et al. 2019b), reinforcement learning (RL, e.g., Zoph and Le 2017; Zoph et al. 2018), and gradient-based algorithms (e.g., Liu et al. 2019; Pham et al. 2018).

Within the BO framework, BANANAS (White et al., 2019) has emerged as one state-of-the-art algorithm (White et al., 2019; Siems et al., 2020; Guerrero-Viu et al., 2021; White et al., 2021). The two main components of BANANAS are a (truncated) path encoding, where architectures represented as directed acyclic graphs (DAG) are encoded based on the possible paths through that graph, and an ensemble of feed-forward neural networks as surrogate model. Recently, White et al. (2021) investigated the performance of different surrogate models in the context of BO-based NAS and concluded that the strong performance of BANANAS on the NAS-Bench-101 benchmark (Ying et al., 2019) is determined by its path encoding and not its choice of surrogate model. Results suggest that path encoding leads to a performance boost on smaller search spaces (such as the one of NAS-Bench-101) but does not scale well on larger search spaces such as DARTS (Liu et al., 2019).

We hypothesize that for larger search spaces, the strong performance of BANANAS stems from its choice of acquisition function optimizer in the sense that local optimization of architectures is most important and other components have less impact on performance. To investigate this hypothesis, we vary the main BANANAS components, namely architecture representation, surrogate model, acquisition function and acquisition function optimizer in a

factorial manner and examine the performance difference on the NAS-Bench-301 benchmark (Siems et al., 2020)[1].

## 2. BANANAS

BANANAS (White et al., 2019) uses a (truncated) path encoding, combined with an ensemble of feed-forward neural networks as surrogate model, to predict the performance of architectures. Cell-based search spaces such as DARTS can be encoded by representing cells as DAGs, with nodes as vertices and connections with operations between them as edges. For every *path*, i.e., every possible ordering of vertices, a binary feature is generated, indicating whether the DAG contains all directed edges along this path. If architectures are created by sampling edges in the DAG subject to a maximum edge constraint (i.e., limiting the number of edges), most possible paths have a low probability of occurring (White et al., 2019; Ying et al., 2019). Therefore, BANANAS truncates the least-likely paths, resulting in a relatively informative encoding that scales linearly with the size of the cell.

Let $\mathcal{A}$ denote the search space of architectures and $\{f_m\}_{m=1}^M$ denote an ensemble of $M$ feed-forward neural networks ($\mathtt{NN}$)[2], where $f_m : \mathcal{A} \to \mathbb{R}$. BANANAS uses independent Thompson sampling ($\mathtt{ITS}$, Thompson 1933; White et al. 2019) as acquisition function:

$$\alpha_{\mathrm{ITS}}(x) = \tilde{f}_x(x), \ \ \tilde{f}_x(x) \sim \mathcal{N}(\hat{f}, \hat{\sigma}^2), \tag{1}$$

where $\hat{f} = \frac{1}{M} \sum_{m=1}^M f_m(x)$ and $\hat{\sigma} = \sqrt{\frac{\sum_{m=1}^M (f_m(x) - \hat{f})^2}{M-1}}$. $\alpha_{\mathrm{ITS}}(\cdot)$ is then optimized using the following mutation algorithm ($\mathtt{Mut}$): The best performing architecture so far is selected and mutated in 100 different ways by changing a single operation or edge randomly and the architecture yielding the largest acquisition value is proposed as the next candidate for evaluation.

## 3. Experiments

To investigate the effectiveness of different components of BANANAS on NAS-Bench-301, we conducted a series of experiments where we replaced some of them with what we consider more "standard" choices. A simpler configuration could use a random forest ($\mathtt{RF}$, Breiman 2001; notably used successfully in SMAC, Hutter et al. 2011) as a surrogate model which can either be fitted to path encodings ($\mathtt{Path}$) or natural tabular representations ($\mathtt{Tabular}$) of the architectures as provided in NAS-Bench-301 in the form of a ConfigSpace (see the $\mathtt{ConfigSpace}$ library, Lindauer et al. 2019). In the tabular encoding, architectures are represented by enumerating all nodes and potential edges and introducing categorical hyperparameters for each operation along each potential edge, where the nodes serving as input of each intermediate node are again defined as categorical hyperparameters and operations on a certain edge can only be specified if this edge is actually present in the DAG

---

1. NAS-Bench-301 uses architectures of the DARTS search space trained and evaluated on CIFAR-10 (Krizhevsky, 2009)
2. White et al. (2019) use $M = 5$ sequential fully-connected networks with 10 layers of width 20 by default, initialized with different random weights and trained using permuted training sets, the Adam optimizer with a learning rate of 0.01, and mean absolute error (MAE) loss

(Siems et al., 2020). Note that another possible architecture representation is given by adjacency matrix encoding (Ying et al., 2019; White et al., 2020a), which was not considered by us. Looking at the acquisition function, the expected improvement (`EI`) is a well-known alternative:

$$\alpha_{\mathrm{EI}}(x) = \mathrm{E}_y[\max(y - y_{\max}, 0)], \tag{2}$$

given in Jones et al. (1998), where in our context $y_{\max}$ is the best validation accuracy observed so far and $y$ is the surrogate prediction of architecture $x$. As a very simple alternative, one could also only be interested in the posterior mean prediction (`Const. Mean`) as acquisition function, which does not take the surrogate model uncertainty estimates into account. Finally, looking at acquisition function optimizers, a popular choice is given by random search (`RS`): Drawing a large number of architectures uniformly at random (e.g., by sampling from the ConfigSpace) and selecting the architecture with the largest acquisition value. Our `RS` method samples 1000 architectures in each BO iteration.

## 3.1 Different BANANAS Configurations on NAS-Bench-301

Choices for the architecture encodings, surrogate candidates, acquisition functions, and acquisition function optimizers were crossed in a full factorial manner (where possible), resulting in overall 18 different algorithms. BANANAS, local search (`LS`) and random search (as NAS method, `Random`) were used as implemented in `naszilla` (White et al., 2020a). In `LS` (White et al., 2020b), all neighbors (e.g., all architectures differing in one operation or edge) of an incumbent are evaluated and the incumbent is replaced if a better architecture has been found and the process is repeated until no better architecture can be found (i.e., a local optimum is reached) or another termination criterion is met. Regarding the reference BANANAS implementation, two configurations were used differing in the frequency of updating their ensemble of feed-forward networks ($k = 1$, i.e., after every iteration, or $k = 10$, see White et al. 2019). The initial design for all methods consisted of ten architectures that were sampled uniformly at random (note that `LS` and `Random` do not rely on an initial design and simply start from zero evaluations). All methods were run for 100 iterations (architecture evaluations) and all runs were replicated 20 times. Results are shown in Figure 1, where the validation accuracy is plotted against the batch number. Note that in each facet, the reference `naszilla` implementations of BANANAS, `LS`, and `Random` are provided and by design, `Paths + NN + ITS + Mut` is a (re-)implementation of the BANANAS ($k = 1$) configuration. In general, using `Mut` as acquisition function optimizer always results in a strong performance boost compared to using `RS`. Notably, BANANAS' ensemble of feed-forward neural networks, together with path encoding only performs well if combined with `Mut` and is otherwise outperformed by `Random`. Moreover, the very simple configuration of `Tabular + RF + EI + Mut` performs similarly to the reference BANANAS implementation. Finally, neglecting all uncertainty in the predictions by opting for the `Const. Mean` acquisition function results in very good performance when combined with `Tabular + RF + Mut`. Performing a one-way ANOVA on the top seven algorithms indicated no significant difference in final performance, $F(6, 133) = 1.026, p = 0.411$. Table 1 presents results of a four-way ANOVA on the final performance of the 18 algorithms outlined above with respect to the factors architecture encoding, surrogate candidate, acquisition

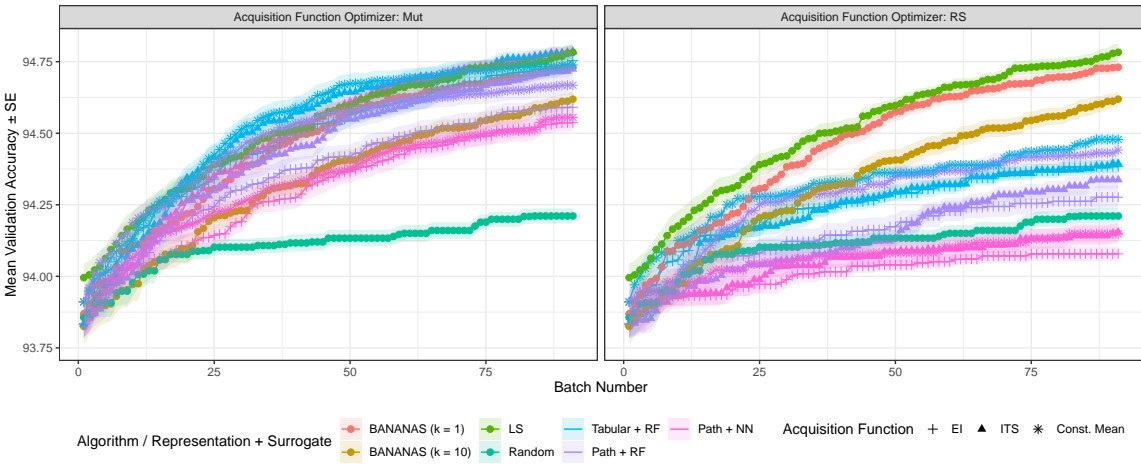

**Figure 1:** Different BANANAS configurations on NAS-Bench-301. Mean validation accuracy with standard error bands, higher is better. Color: optimization method and surrogate model. Facet: acquisition function optimizer, where applicable. Point shape: acquisition function, where applicable. The `ITS` acquisition function and `Mut` acquisition function optimizer is used for BANANAS methods, and `LS` and `Random` do not use an acquisition function; their accuracy is therefore shown in both facets of the graph.

function, and acquisition function optimizer. The acquisition function optimizer is by far the most important determinant of final performance.

| | Sum Sq | Df | F value | Pr(>F) |
|---|---|---|---|---|
| Architecture Encoding | 0.41 | 1 | 19.57 | 0.0000 |
| Surrogate Candidate | 1.01 | 1 | 48.31 | 0.0000 |
| Acquisition Function | 0.56 | 2 | 13.49 | 0.0000 |
| Acq. F. Optimizer | 13.18 | 1 | 632.43 | 0.0000 |
| Residuals | 7.38 | 354 | | |

**Table 1:** Results of a four-way ANOVA on the factors architecture encoding, surrogate candidate, acquisition function, and acquisition function optimizer. Type II sums of squares.

## 3.2 Examining the Effect of the Acquisition Function Optimizer

To investigate the performance difference with respect to the acquisition function optimizers, another experiment was conducted. Based on the `Tabular + RF + EI` configuration three different acquisition function optimizers were compared: Random search with 100000 architectures drawn uniformly at random in each BO iteration (`RS+`), random search as described above (`RS`) and `Mut` as described above. Ten architectures were sampled uniformly at random and used as the initial design points for all replications. All methods were run for 100 iterations (architecture evaluations) and all runs were replicated 20 times. Results are given in Figure 2A. As can be seen, `Mut` strongly outperforms even the `RS+` optimizer.

We collected additional data in the `RS+` runs shown in Figure 2A. In each BO iteration of these runs, we also performed acquisition function optimization using the other two methods (`RS` and `Mut`) and investigated the properties of the proposed architectures. While the op-

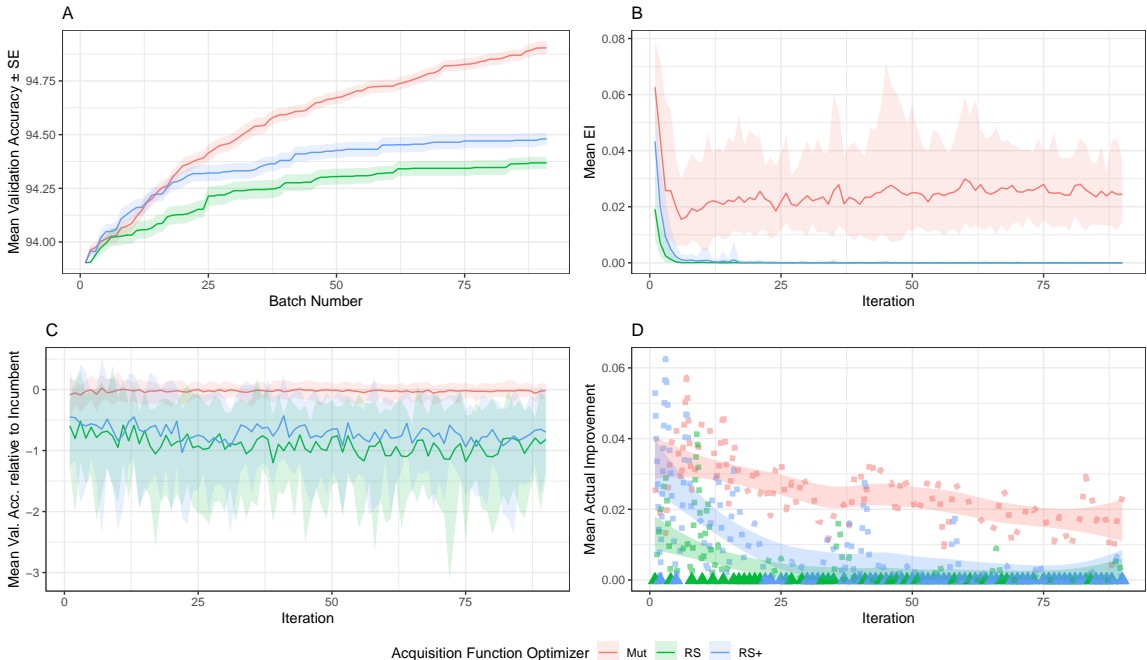

Figure 2: `Tabular + RF + EI` with different acquisition function optimizers on NAS-Bench-301. A: Validation accuracy. B: `EI`. C: Validation accuracy relative to the incumbent. D: Actual improvement. Ribbons in B and C represent 2.5% and 97.5% quantiles. In D, LOESS smoothing was performed and triangles indicate no improvement.

timization itself proceeded with the architectures proposed by `RS+`, the collected data gives information about the quality of architecture proposals done by the other methods. The data collected was the `EI` of each proposed architecture, according to the surrogate model (Figure 2B), the actual validation accuracy of each proposed architecture (when evaluated), minus the validation accuracy of the incumbent during that iteration (Figure 2C), and that same quantity, conditional on the proposed architecture giving higher validation accuracy than the incumbent ("actual improvement", Figure 2D).

`Mut` results in both higher `EI` and actual improvement, i.e., `Mut` solves the inner optimization problem better than the other optimizers and the actual improvement is comparably large. Note that the difference between the validation accuracy of proposed architectures and incumbent is mostly negative due to a fixed iteration seldom resulting in actual improvement. Looking at Figure 2D, we observe that following the proposals by `RS+` and `RS` results in many iterations with no improvement (as indicated by triangles).

In a final experiment, focus was given to the accuracy of the surrogate model when predicting the validation accuracy of architectures depending on the edit distance to the incumbent. Based on the `Tabular + RF + EI + Mut` configuration, the BO loop was run for 50 iterations (architecture evaluations); the construction of the initial design remained the same and all runs were replicated 100 times. For edit distances ranging from 1 to 8, 100 test architectures were constructed each by mutating a fixed number of parameters (operations or edges) of the incumbent. For these test architectures, Kendall's $\tau$ with respect to the predicted and true validation accuracy (after evaluation) is given in Figure 3A.

Additionally, the true validation accuracy is plotted against the edit distance (Figure 3B), with the gray point representing the incumbent. In Figure 3C, the expected improvement and the actual improvement is plotted. While the true validation accuracy decreases when increasing the edit distance, Kendall's $\tau$ increases, suggesting that the surrogate model is not capable of precise performance prediction for high performing architectures close to the incumbent. This finding goes in line with results of White et al. (2021) that model based NAS methods perform bad when predicting the performance of neighbors of high performing architectures when the search space is large. Moreover, the expected improvement is relatively unaffected by the edit distance, although the actual improvement is largest for close architectures. This may indicate that thorough optimization of the acquisition function is not needed, instead simply considering neighboring architectures as candidates may be sufficient.

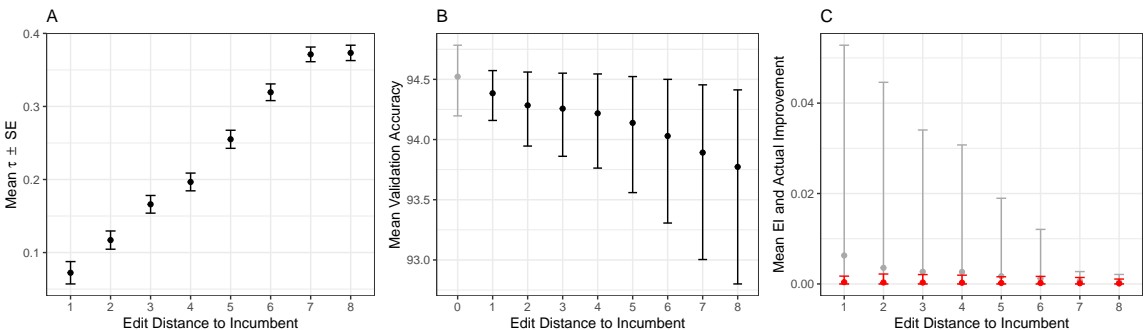

Figure 3: `Tabular + RF + EI + Mut` on NAS-Bench-301. A: Kendall's $\tau$ of the predicted and true validation accuracy of test architectures constructed to have different edit distances to the incumbent. B: True validation accuracy of these test architectures. Validation accuracy of the incumbent is given in gray. C: Expected Improvement (red) and actual improvement (gray) of these test architectures. Bars in B and C represent 2.5% and 97.5% quantiles.

## 4. Discussion

We have presented empirical results suggesting that the performance of BANANAS on large cell-based search spaces such as DARTS is predominantly determined by its choice of acquisition function optimizer that is effectively performing a randomized local search. Other components such as the architecture encoding, surrogate model and acquisition function have a comparably small effect on the performance, and exchanging most components of BANANAS with more "standard" choices results in a method that is not significantly worse. Local search, which uses no surrogate model at all, does in fact perform equally well (at least on the NAS-Bench-301 benchmark), giving more evidence that the local nature of BANANAS' mutation acquisition function optimization contributes mainly to its success. Minimally mutating the incumbent allows for solving the inner acquisition function optimization problem better than random search variants with large budget, although the surrogate model suffers from imprecise surrogate predictions for architectures close in edit distance to the incumbent. Future work on BO methods for NAS should therefore also focus on algorithms for solving the inner acquisition function optimization problem.

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

## Appendix A. Computational Details

The BO algorithms were implemented in R (R Core Team, 2020) within the `mlr3` (Lang et al., 2019) ecosystem relying on `mlr3mbo` (version 0.0.0.9999; Richter et al. 2021) and `bbotk` (version 0.3.0.9999; Becker et al. 2021). Random forests were used as implemented in the `mlr3extralearners` package wrapping `ranger::ranger` (version 0.12.1; Wright and Ziegler 2017) with `num.trees` set to 500, `se.method` set to `"jack"`, and `respect.unordered.factors` set to `"order"`. Missing values were encoded with a new level ".missing" via a preprocessing pipeline built using `mlr3pipelines` (version 0.3.0; Binder et al. 2020).

Python 3.8.7 was used via the `reticulate` package (version 1.18; Ushey et al. 2020) within R. For NAS-Bench-301, `nasbench301` version 0.2 (Siems et al., 2020) was used relying on the `xgb_v1.0` surrogate model for the validation accuracy. The feed-forward ensemble of neural networks and path encoding as used by BANANAS was directly adopted as implemented in `naszilla` (version 1.0; White et al. 2020a). BANANAS, local search and random search (as NAS methods) were run using `naszilla` employing the same `nasbench301` setup as described above under Python 3.6.12 (due to different module requirements).

All computations were performed on 2 Intel© Xeon© E5-2650 v2 @ 2.60GHz CPUs each with 16 threads using R 4.0.3 under Ubuntu 20.04.1 LTS. Parallelization in R was done via the `future` (Bengtsson, 2020) and `future.apply` (Bengtsson, 2020) packages (version 1.21.0 and 1.7.0) on top of the internal parallelization of the `data.table` (Dowle and Srinivasan, 2021) package (version 1.14.0).

## Appendix B. NAS Best Practices Checklist

Here, we answer to applicable questions of the NAS best practices checklist (version 1.0), see Lindauer and Hutter (2019).

- as NAS benchmark, NAS-Bench-301 (`nasbench301`) version 0.2 was used relying on the `xgb_v1.0` surrogate model (deterministic) for the validation accuracy

- all computations were run on the same hardware (2 Intel© Xeon© E5-2650 v2 @ 2.60GHz CPUs)

- all results reported are based on ablation studies

- the same evaluation protocol was used for all methods

- performance was compared with respect to the number of architecture evaluations

- random search was included as a NAS method

- multiple runs (20 or 100) were conducted; reproducibility with respect to algorithms implemented in R is given due to an initial random seed being set; regarding `naszilla`, no seed can be explicitly set

