# OpenReview forum: "Mutation is all you need"
_ICML.cc/2021/Workshop/AutoML — AutoML@ICML2021 Poster_

### Official Review · Reviewer_Bm1E · 2021-06-08
**Review:**

**Rating:** 6
**Confidence:** 5

**Review:**

The issue with this paper is that it is incredibly hard to follow and that I think the abstract introduction and title are actually a very bad representation of what makes the results interesting. The score I gave is marginally above the acceptance threshold. If it were a regular conference submission I would actually lower the score to rejection since there are so many points not clear.

What the paper shows is that when you use a value function to predict performance, trying to cover the entire search space is hard. But focussing on a subset of the search space by only evaluating close to models you actually have training data for actually works well.

- The description of the path encoding is incredibly confusing. (But it is so too in the Bananas paper and the follow up work done by the same authors). I struggled for quite some time before I realized this would only work with small cell based search spaces and not with search spaces as used in for example proxylessnas. It might be useful to stress this point because this also explains that very long path are unlikely, which leads to the compression.

- When local search is defined on page 3, it is not clear whether all models in the neighborhood are actually evaluated using training or using a value function. THis only becomes clear on page 6.

- Given the fact that for local search the neighbors are actually trained, I think that Fig. 1 Might actually be misleading. The x-label says batch number, but local search evaluated many more models with full training compared to some of the other methods.

- Properly understanding what method does what is hard. All methods in Fig. 1 need to be described in a more structured way. e.g.
** Local search: description
** Random: description
** Bananas: description
** Variations: describe how they vary,

- The const mean idea is not described properly. I assume this is similar to the independent tompson sampling but with 0 variance?

- It is often not clear how many models were used for training the value functions.

- Figure 1 is way too dense and it is impossible to make sense of. (Even more so for the poor people that have to print on US Letter paper, which is smaller than the A4 paper used presumably by the authors).


Finally, the title is simply a bad idea. The title "Attention is all you need" was already ridiculous on one of the most successful papers in machine learning. "Mutation is all you need" is certainly not an appropriate summary of the work in this paper.
-  The work is evaluated on a single artificial benchmark. While benchmarks are useful for comparison, making the claim that the mutation is all you need would require a much bigger evaluation. There is no evidence provided that this actually results in better improvements for NAS in general.
- I do not think that this title is actually beneficial to the authors. It was definitely not obvious to me that this was a NAS paper and for that reason I would ignore it if I saw it listed in some (general, non-workshop) conference proceedings.

---

### Official Review · Reviewer_XriL · 2021-06-17
**Review for Mutation is all you need**

**Rating:** 7
**Confidence:** 5

**Review:**

[Work Significance]
This work studies different BANANAS configurations for larger search space (NAS-Bench-301). The author[s] categorized it into four different parts and did combinatorial experiments to study which part gives the most influence on the overall search performance. To this end, they conclude that the acquisition function is the most important determinant of the final performance, and mutation outperforms random search in the inner acquisition function optimization.


[Something that could be improved]
Mutation as the acquisition optimizer makes more sense to me compared to random sampling, since the surrogate model already predicted the promising ones. Within the mutation function, it would be better to do an even more comprehensive ablation on the combination of 1). the number of architectures N used for the mutation, 2). the max edit distance D for each,  3). how many architectures M to sample for each edit distance, and 4). how many architectures K with the highest acquisition score is chosen for the full evaluation. This results in choosing K from a total of N * D * M combinations, which could give a significant statistical difference across different search spaces compared to Figure 3 (c).

The summary of encodings is not comprehensive enough (e.g. continuous, learning-based), I would suggest https://arxiv.org/pdf/2006.06936.pdf, https://arxiv.org/pdf/2102.07108.pdf for a reference. It would be very interesting to see how different mutation schemes can be adapted to continuous vectors and their corresponding performance, e.g. discretize them into the fixed number of bins of the same size and then do the mutation, or make the bins adaptive.

---

### Official Review · Reviewer_SDgs · 2021-06-18
**Small contribution towards understanding of the performance of BANANAS**

**Rating:** 5
**Confidence:** 4

**Review:**

This paper investigates the importance of different components of BANANAS for good performance on neural architecture search tasks.  BANANAS is just one of many Bayesian approaches for neural architecture search (NAS), therefore, the scope and impact of the paper is fairly limited.  Perhaps a more interesting comparison would be an evaluation of multiple Bayesian NAS methods to identify the necessary components for good performance and/or propose a combined method that outperforms all existing approaches.

The paper performs ablation studies to capture the performance attribution of four components of BANANAS: search method, surrogate function, acquisition function, and acquisition optimization function.  Of these four, the acquisition optimization function has the highest impact on performance and in particular, mutation-based optimization drastically outperforms random.  This result by itself seems unsurprising as local search is known to work well for NAS; the statement would carry more weight if mutation-based acquisition function optimization was the key to the outsized performance of multiple Bayesian methods.

Pros:
- Clearly written.

Neutral:
- Experiments are thorough but results only provided for NAS-Bench-301.

Cons:
- The paper as it stands is limited in scope and impact.
- The topic addressed in the paper limits its novelty.

---

### Decision · Program_Chairs · 2021-06-21

Accept (Poster)